# Comparative Analysis of Usability and Accessibility of Kiosks for People with Disabilities

Yuryeon Lee [1], Sunyoung Park [1], Jaehyun Park [2] and Hyun K. Kim [1,3,*]

1. Department of Artificial Intelligence Convergence, Kwangwoon University, Seoul 01897, Republic of Korea
2. Department of Industrial and Management Engineering, Incheon National University (INU), Incheon 22012, Republic of Korea
3. School of Information Convergence, Kwangwoon University, Seoul 01897, Republic of Korea
* Correspondence: hyunkkim@kw.ac.kr

**Abstract:** Owing to technological advancements, kiosks have become more prevalent in public places. When using such kiosks, elderly persons and people with disabilities face problems related to accessibility and usability, such as difficulties in kiosk operations such as menu selection and in accessing the kiosk space. Previous studies have usually included accessibility as a subset of usability. However, in this study, we aim to redefine the relationship between these two concepts with a focus on newly emerging kiosk devices. First, we performed a literature review to thoroughly analyze these concepts. Then, we conducted a focus group interview (FGI) targeting people with visual, hearing, and physical impairments to learn about the difficulties that these people face when using kiosks. Finally, we analyzed the characteristics of accessibility and usability related to kiosks and designed a diagram that illustrated the relationship between them. While accessibility and usability shared similarities regarding consistency and user control, they differed deeply regarding their subcategory items; many opinions on accessibility were related to essential functions, whereas many on usability were related to psychological factors such as additional functions or personal preferences. These results can be useful when creating laws and guidelines regarding the accessibility and usability of kiosks or when developing kiosk functions.

**Keywords:** kiosk; self-service terminal; accessibility; usability; disability



## 1. Introduction

With the technological developments made since the Fourth Industrial Revolution, devices such as smartphones and personal computers (PCs) have rapidly grown in number [1]. Kiosks, which are unmanned terminals that allow users to obtain information through information services or task automation without any human assistance [1–3], are natural and intuitive information systems that have become an important element of the information age. Each kiosk supports multimedia (audio, video, photos, and animation) functionalities, allowing users to interact with them in different ways [4]. Public kiosk systems are used for various purposes, including taking pictures, connecting devices to the Internet, purchasing tickets, obtaining financial and administrative services, and learning directions. The service industry has been surging with the introduction of kiosks, and developments in touch screen technology have enabled their easy accessibility in daily life [5]. These devices have become common in various public places such as restaurants, subways, museums, shopping malls, hospitals, movie theaters, libraries, and airports, and they are gradually growing in number [1,4,6–8].

Recently, artificial intelligence (AI) concierges with technologies such as facial and voice recognition and thermal imaging cameras have been introduced to solve the accessibility and usability problems that elderly persons face when using kiosks. Such concierges welcome and guide customers and deliver basic information regarding topics such as

exchange rates and the weather. AI concierges in the financial sector use a large screen attached to the kiosk and guide users by a simulated natural conversation. Regarding the advantages on the provider side, kiosks can continually provide a high quality of service and reduce labor costs as they are operable 24 h a day; on the consumer side, customers can benefit from the services quickly and not have to wait in line for a long time [9]. However, even such generally used kiosks still pose problems regarding accessibility and usability for people with disabilities and for the elderly [10,11]. When using a kiosk, not only are difficulties experienced in operating procedures such as menu selection, but also in accessing the kiosk space itself. If such accessibility issues are not considered when designing and using a kiosk, there will be severe restrictions on the daily use of kiosks by such people [10,12].

Usability is generally defined as an individual's ability to perform a task [13–15]. The International Organization for Standardization (ISO) 9241-11 defines usability as "the extent to which a specific user can use a product to achieve a specific goal with effectiveness, efficiency, and satisfaction in a specific use environment" [16]. Usability is an important element of the existing user experience (UX) and includes items such as simplicity, directness, efficiency, informativeness, flexibility, learnability, and user support, which measures how easily and conveniently a product/service can be used [17,18]. According to the ISO, accessibility is defined as the ease of use of a product, service, environment, or facility, regardless of individuals' capabilities [19]. Products/systems with accessible designs are ones that users with disabilities can use [20]; such a design is referred to as inclusive design [21] and universal design [22]. Although accessibility and usability are different concepts, they are sometimes used interchangeably without clear differences in existing studies [23]. Their ISO definitions make them seem similar, and they seem to only differ with respect to how users with disabilities perceive them. Accessibility mainly refers to the capability of an individual to access a specific device, and its definition has been covered in many legislations [5,24,25]. Regarding kiosk devices, there are various existing legislations, and new ones are also being enacted [5]. In the United States, kiosk-related legislations exist in Section 508 of the Rehabilitation Act, US Air Carrier Access Act, Americans with Disabilities Act Standards for Accessible Design, and kiosk-related legislation has also been recently added to the European Accessibility Act. In Korea, the Guidelines for Public Access Terminal Accessibility Standard Document exists; this documentation was included in the Digital Divide Reduction Act in 2021. In contrast, usability has been considered in the development of design interface guidelines along with UX concepts [26]. Despite usability being already somewhat established in the field of technology, some of its characteristics have been vaguely defined; it has sometimes been used interchangeably with terms such as "dummy proofing" and "user-friendliness", which reduces the importance of its meaning in interaction design and negatively affects how user performance is viewed in terms of productivity. Furthermore, visual design, which is an essential element of usability that is commonly misunderstood with visual appeal, is not the only part of interaction design [27]. There have been studies on kiosk usability that have included evaluations of the usability of kiosk systems specialized for wayfinding [4,6] as well as studies on the user interface (UI) design guidelines of kiosks [28,29]. However, there is insufficient research on the design of new kiosk UIs with a focus on user needs [30].

The purpose of this study is to systematically analyze the concepts of accessibility and usability with a focus on kiosk devices. Even if a specific product satisfies the required accessibility requirements, it does not mean that users with disabilities have a satisfactory UX when using it [24]. Accessibility needs to be considered in the development stage of the product/device and during the enactment of the relevant laws; using certain measures, the compliance of a specific/product service with laws can be confirmed [10]. Regarding usability, its inclusion in legislation needs to be carefully considered because there are many changes with the related laws conflicting with the freedom and creativity of a company's product/device design. Particularly regarding legal regulations, many cases trail behind technological developments [10]; therefore, when applying or creating new technology, it is necessary to focus on accessibility separately from usability. As mentioned earlier,

there are many studies that individually define accessibility and usability but few that systematically compare and analyze them. Furthermore, the studies that have compared and analyzed these concepts mainly involved websites and smartphones. In addition to kiosks being used for obtaining information, there are other situations wherein they are used, including in facilitating a payment service using a credit card/smartphone or a discount service on a welfare card for people with disabilities. Therefore, it is necessary to redefine the accessibility and usability of kiosks while considering these circumstances. To this end, the concepts and characteristics of accessibility and usability were comprehensively investigated through a literature review, and focus group interviews (FGIs) were conducted, targeting individuals with visual, hearing, or physical impairments. Through these interviews, a clear concept regarding the accessibility and usability of a kiosk could be defined. Based on our assessments, we have presented a diagram defining the concepts of accessibility and usability, which is expected to be useful when designing kiosk features while considering accessibility and usability in the future or when creating legal and practical design guidelines for kiosks.

## 2. Literature Review

### 2.1. Accessibility and Usability

Many studies have conducted intensive literature reviews to compare and analyze the concepts of accessibility and usability. Thatcher et al. [31] defined accessibility as a subset of usability that only affects people with disabilities while defining usability as a concept that affects both people with and without disabilities. On the other hand, Petrie and Kheir [32] subdivided the concepts of pure accessibility, pure usability, and universal usability, targeting both people with and without disabilities. Pure accessibility concerns people with disabilities, pure usability only concerns those without disabilities, and universal usability concerns both those with and without disabilities. In their study, the types of problems experienced by both people without disabilities and visually-impaired people were identified on a website, and the relational characteristics between these problems were identified. Accordingly, Thatcher et al. derived problems related to universal usability for both people with and without disabilities, which contrasts with the concept in the study. The meaning of usability, UX, and accessibility and the relationship between these concepts have been investigated and analyzed [33]. Accessibility is defined as a concept that does not include affective elements such as fun and enjoyment, and it is closer in meaning to usability than it is to UX. When people with disabilities interact with a device, their UX includes many emotional elements as well as usability [34]; thus, it is necessary to distinguish UX from accessibility. Aizpurua, Harper, and Vigo [35] investigated the relationship between UX properties and web accessibility. They measured the accessibility in terms of that perceived by the participants and that specified in the guidelines. The results showed that the web accessibility perceived by the participant had a significant correlation with 27 out of 35 UX attributes. Petrie and Bevan [23] introduced the concepts of usability, accessibility, and UX as criteria for developers to evaluate the E-system. Despite the clear consensus standard, the relationship between accessibility and usability was still unclear. In this study, the difficulties experienced by people with disabilities and by the elderly when using the E-system were defined as accessibility issues, and the difficulties experienced by people with disabilities and younger users were defined as usability issues.

In previous studies, the usability of a product/service for a person with a disability was defined as accessibility [31,32], and accessibility was mainly studied as a sub-item of usability [36]. However, as discussed in the existing literature [32], the concept of universal usability exists, and people with disabilities also perceive usability when using products/services; therefore, the concepts of accessibility and usability need to be separately defined with respect to the needs of people with disabilities. In this study, the concepts of accessibility and usability for people with disabilities were defined as follows. Accessibility indicates the degree of ease for people with disabilities to access the main functions of devices/services [37] and should be considered essential when using devices/services.

Usability for people with visual impairments can also be considered "usable accessibility" [38], which indicates how effectively and efficiently a device/service can be used [32]. It mainly includes considerations regarding the psychological discomfort or convenience of a person in using the device. Based on these conceptual characteristics, accessibility should be prioritized over usability for users with disabilities. Users with disabilities cannot use a device/service regardless of the level of usability if accessibility has not been guaranteed.

## 2.2. Methods for Evaluating Accessibility and Usability

In several studies, various methodologies have been used to evaluate the accessibility and usability of websites and mobile systems. In this study [39], the usability of a website was evaluated for users with visual impairments; accordingly, guidelines considering accessibility and usability were developed to help users that used screen readers to assist them in accessing the website. A methodology to evaluate the accessibility and usability of mobile computing applications has been developed [40]. First, an accessibility guideline was analyzed considering the specificity of the mobile system. Then, usability heuristics were introduced, and the results of applying the developed methodology to mobile computing applications were summarized.

Many studies have identified problems in the accessibility and usability of websites for the elderly and people with disabilities [41–44]. Evaluation methods have been established for startups and mobile games; mobile game tests have also been conducted for the elderly [45]. Through these tests, issues regarding usability, accessibility, and UX were identified, and guidelines on usability and accessibility heuristics were accordingly provided. Even with these guidelines, the users faced problems, including taking a long time to understand the interface elements or incorrectly touching them. In this study, there are no specific provisions like those under TAW (tool for analyzing website accessibility) introduced by the Technological Center for Information and Communication Foundation (CTIC) or the Web Content Accessibility Guidelines (WCAG) that considers the motor, cognitive, and visual impairments associated with aging in the usability and accessibility guidelines for website design [46]; obtaining such guidelines might be difficult. To improve the interaction, readability, and usability of the content, the information regarding the UI and design items in the accessibility and usability guidelines was reviewed. Alajarmeh [47] conducted a study to evaluate the accessibility of public health websites from 25 countries using WCAG 2.0.

## 2.3. Study on Improvement in Kiosk Accessibility

There are relatively few papers on the accessibility and usability of kiosk devices compared with those on websites or mobile applications. A checklist has been developed to supplement and improve the Information and Communication Accessibility Guidelines for the Disabled and Aged [36]. Additionally, a method has been proposed to quantitatively evaluate accessibility guidelines. The accessibility of public institution kiosks was surveyed; the results revealed that public institution kiosks had a low level of accessibility. The variation in individual kiosks for each checklist item was quite large, which was attributed to differences in the design methods and management for the products. Sahua and Moquillaza [48] systematically reviewed the interface of an automated teller machine (ATM) and a methodology for evaluating the usability of tools. The authors selected 12 studies out of 132, and the most used tools and their corresponding frequencies were identified in the selected studies for the usability evaluation. A usability evaluation has been conducted on the interface of a railway self-service terminal based on tests and interviews regarding user performance [49]. The results showed that the elderly had many usability problems with self-service terminals. To solve these problems, it has been recommended that work steps be minimized, appropriate feedback be provided, and a simple interface be designed. Laws and guidelines for the accessibility of unmanned information terminals have been investigated, and the characteristics of each guideline were compared and analyzed based on three types of disabilities and seven accessibility functions [5]. The disability rate for

each guideline and accessibility function and the guideline rate for each accessibility function were identified. Accessibility guidelines such as Section 508, the Twenty-First Century Communications and Video Accessibility Act (CVAA), European Accessibility Act (EAA), and WCAG have been analyzed from a UX perspective; additionally, guidelines for developers and designers of IT devices have been analyzed from a UX perspective, and aspects to be considered when developing access guidelines have also been previously presented [25]. Tüzün, Telli, and Alır [4] performed a usability test based on a kiosk with a 3D touch screen in a shopping mall in Turkey; six wayfinding tasks consisting of a shortcut menu, category menu, and search menu were provided to the participant, and the menu selected to perform each task was accordingly identified. The results revealed that most participants failed Task 6 and spent the least amount of time in Task 5. Maguire [28] reviewed guidelines for the design of the UI of kiosk systems based on previous studies. Guidelines for kiosk evaluation were provided based on 22 factors, which included physical access, privacy, structure, and navigation. Sandnes et al. [30] evaluated the UX of high-speed train ticket vending machines in Taiwan. Such a ticket vending machine was evaluated based on the UI design heuristics for public kiosks, and several improvements were proposed. After identifying problems such as language version revision, color, and location in the derived improvements, a UI design heuristic was proposed based on the improvements. The user experience design (UXD) for a kiosk in the Istanbul Public Transport System (IPTS) was presented in [7]. First, a survey was conducted based on methods such as user research and personas, heuristics, and UX tests to derive the main considerations falling under the UXD. Based on this, a usability test was conducted on the prototype, and major improvements were identified. Additionally, the UXD approach was verified using various methods.

## 3. Method

### 3.1. Participants

Overall, we conducted FGIs with 31 users (Figure 1). The participants were recruited from among users with visual, hearing, and physical disabilities who had experience using a kiosk or were interested in using one. Each FGI included two groups for people with each type of impairment (visual, hearing, and physical) over six sessions. Regarding the sessions for people with visual impairments, each session consisted of (1) five completely visually-impaired people using only screen readers and (2) six people with low vision who used only screen readers, residual vision, or both, depending on individual preferences. For the sessions with people with hearing impairments, there were (1) five people who communicated via sign language, (2) one person who communicated orally, and (3) four people who communicated via both. Finally, for the sessions with people with physical impairments, there were (1) five people with mild disabilities and (2) five with spinal cord-related disabilities.

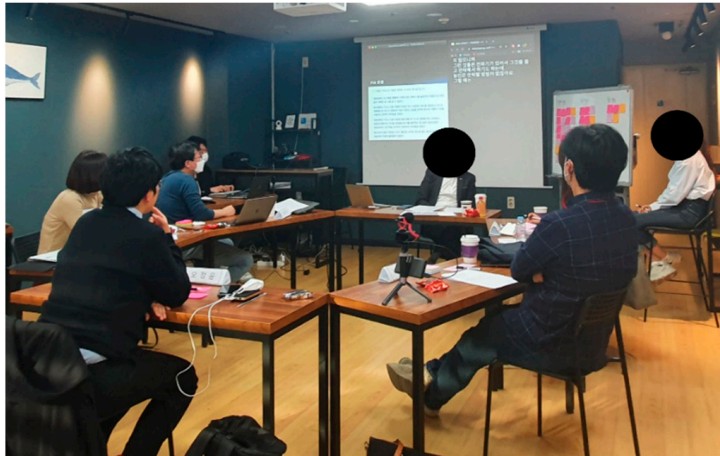

**Figure 1.** Example of a focus group interview (FGI) (session for people with hearing impairments).

*3.2. FGI Procedures*

Each session in the FGI lasted an average of 2 h and was overseen by a main facilitator and three assistant facilitators. The main facilitator explained the questions and allowed the participants to freely share their opinions, and the assistant facilitators took notes on the interviews in shorthand and also took pictures and videos. In sessions involving participants with hearing impairments, a hired professional stenographer noted down the opinions of the participants and displayed them on the screen in real time, as shown in Figure 1. Additionally, the main facilitator communicated with the participants through sign language to facilitate their understanding. In sessions involving visually-impaired participants, questionnaires prepared using Google Forms were delivered before and after the interview for screen reader use. Through a pre-survey, we investigated the sex, affiliation, and status of the participants, the main use areas, and the frequency of use of the kiosks. Then, a survey was conducted before and after the interviews to collect data on the actual conditions under which the unmanned information terminals were used. Opinions related to the need for the enactment of related laws, satisfaction with use, preferred access, and areas desired for use were collected. This study was approved by the Institutional Review Board of Kwangwoon University (IRB approval: 2020-08-005).

*3.3. Data Analysis Procedure*

The opinions derived from the FGIs were classified into accessibility and usability items. Then, referring to previous studies, mapping was performed according to individual criteria.

First, the study conducted by Park et al. [16] was referenced for selecting the subfactors of usability and accessibility. The usability-related items were classified into 7 primary categories and 20 subcategories based on the usability items in this study. The primary categories related to the items were divided into simplicity, directness, efficiency, informativeness, flexibility, learnability, and user support items; the subcategories are shown in Table 1.

**Table 1.** Classification of usability.

| Primary Category | Subcategory |
| --- | --- |
| Simplicity | Modelessness |
| Directness | User control |
| Efficiency | Effectiveness, Effortlessness |
| Informativeness | Comprehensiveness, Explicitness, Visibility, Legibility/Readability, Consistency |
| Flexibility | Adaptability, Interoperability |
| Learnability | Memorability, Familiarity, Predictability, Intuitiveness |
| User Support | Easy installation, Error prevention, Forgiveness, Feedback, Helpfulness |

Accessibility was reconstructed using the primary categories in the study of Park et al. [17] and the UI function in the study of Lee et al. [5]; then, it was classified into 4 primary categories and 11 subcategories. The primary categories included directness, informativeness, user support, and efficiency; the subcategories are listed in Table 2.

**Table 2.** Classification of accessibility.

| Primary Category | Subcategory |
| --- | --- |
| Directness | Physical accessibility, User control |
| Informativeness | Physical visibility, Visibility, Legibility/Readability, Consistency |
| User Support | Privacy, Multimodal information, Feedback, Forgiveness |
| Efficiency | Effortlessness |

This qualitative classification was conducted by four accessibility experts, who individually categorized participants' opinions into the categories and subcategories of usability

and accessibility. The principle of classification was that all four experts mapped to the same categories or subcategories; if even one expert had opinions mapped to different categories or subcategories, a discussion was conducted so that all four experts could derive the same classification result.

## 4. Results

Overall, 94 usability opinions were derived through FGIs; these opinions comprised 28, 54, and 12 opinions from individuals with visual, hearing, and physical impairments, respectively. Similar or overlapping opinions from participants with each type of disability were grouped, resulting in 45 items being derived. These items included 15, 22, and 8 types of opinions from participants with visual, hearing, and physical impairments, respectively.

Regarding accessibility, 180 opinions were obtained; these comprised 89, 36, and 55 opinions from individuals with visual, hearing, and physical impairments, respectively. Similar or overlapping opinions from participants with each type of disability were grouped, resulting in 33 items being derived. These items included 21, 3, and 9 types of opinions from participants with visual, hearing, and physical impairments, respectively.

When calculating the number of opinions, duplicate opinions were also obtained. Among the derived opinions, those judged to be ambiguous or inconsistent with the concept of accessibility and usability were excluded. For example, the opinion "BF should be considered first" was excluded because it was considered too comprehensive. Overall, six opinions, including duplicates, were excluded.

### 4.1. Number of Accessibility and Usability Items by Type of Impairment

All of the opinions collected through the FGIs were grouped according to their similarities, organized, and classified overall into 78 items; we classified 33 and 45 items as related to accessibility and usability, respectively. Overall, the items that were related to visual and physical impairments were derived the most and least, respectively (Figure 2). Accessibility was derived in the order of visual, physical, and hearing impairments, and usability was derived in the order of hearing, visual, and physical impairments.

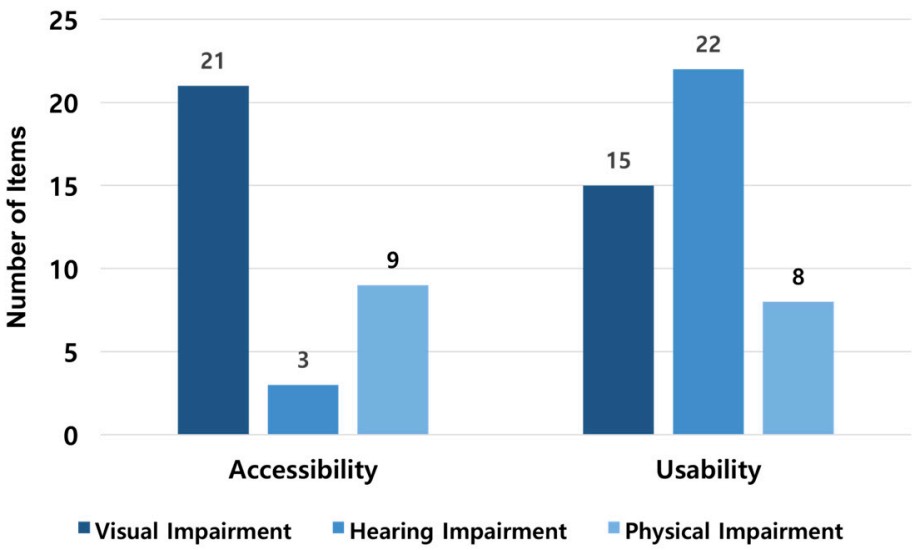

**Figure 2.** Number of accessibility and usability items by type of impairment.

In the subcategories, the opinions related to accessibility were derived in the order of physical accessibility, multimodal information, and user control; those related to usability were derived in the order of effectiveness, helpfulness, and adaptability (Figures 3 and 4).

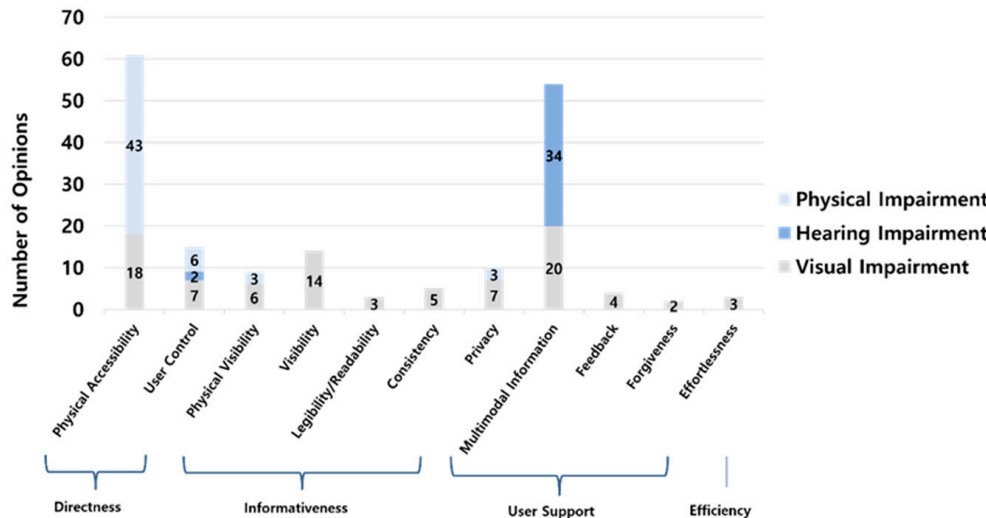

**Figure 3.** Number of opinions according to subcategories of accessibility.

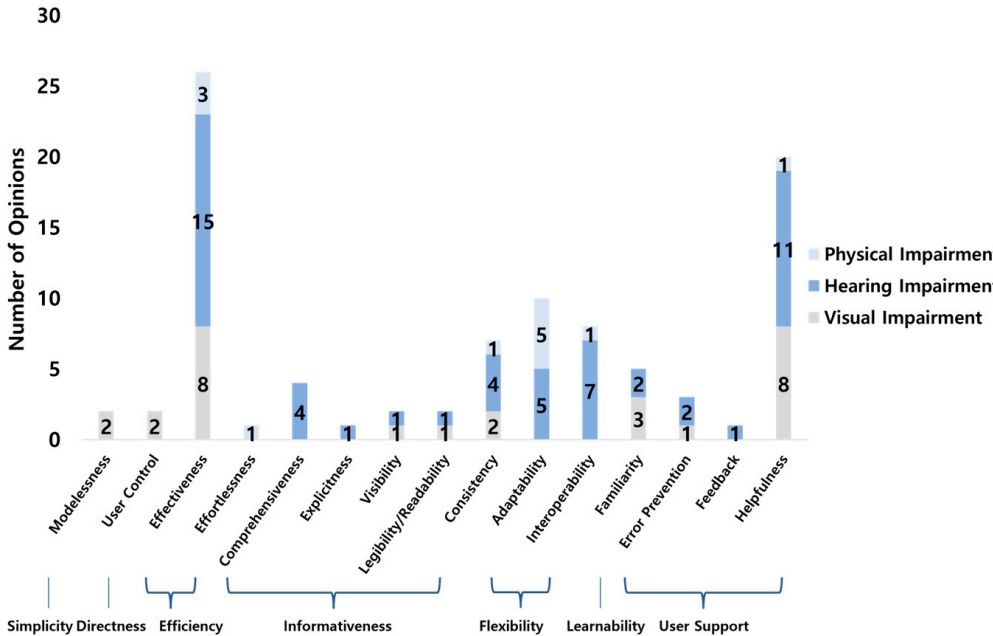

**Figure 4.** Number of opinions based on subcategories of usability.

### 4.2. Analysis of Accessibility Items by Type of Impairment

Participants with different impairments provided many comments on their experience regarding the accessibility of kiosks: participants with visual impairments provided comments such as "providing voice support", "including a high-contrast function", "adjusting kiosk height", and "maintaining consistency in the design of the card slot" (Table 3); those with hearing impairments provided opinions such as "including non-vocal feedback" and "providing sign language interpreters when help button is pressed" (Table 4); and those with physical disabilities provided comments such as "equipping kiosks with wheelchair accessibility", "adjusting kiosk height", and "adjusting kiosk UI" (Table 5).

**Table 3.** Accessibility items for participants with visual impairments.

| Primary Category | Subcategory | Representative Scenario | Number | Example |
|---|---|---|---|---|
| User support | Multimodal information | Providing voice support | 12 | "Since voice support is rarely available, I hope to receive such services" |
| | | Adjusting volume and speed of voice | 3 | "The voice speed needs to be adjusted" |
| | | Implementing a screen reader | 3 | "I need the support of a screen reader to read the information in the middle or listen to the information I want" |
| | | Including a user identification method | 2 | "An alternative to biometric recognition is essential" |
| | Privacy | Providing privacy protection | 5 | "An earphone is needed in the self-service terminal to protect the privacy of the user" |
| | | Requiring security access for ethical and personal reasons | 2 | "Accessibility is needed for both ethical and personal security reasons" |
| | Feedback | Providing error-related feedback | 4 | "Feedback should be provided to help rectify any errors" |
| | Forgiveness | Enabling easy initialization | 1 | "The kiosk should be easy to initialize" |
| | | Including a "previous step" button | 1 | "The lack of a button to return to the previous step is inconvenient" |
| Informativeness | Visibility | Including a high-contrast function | 9 | "I wish the kiosk had a high-contrast mode" |
| | | Providing a physical keypad | 5 | "The numeric keypad needs to be displayed in Braille" |
| | Consistency | Maintaining consistency in the UI | 5 | "I am struggling to learn the operation of this new UI because it is inconsistent" |
| | Physical visibility | Preventing blurring by improving visibility of brightness adjustment features | 3 | "The option to adjust the brightness is difficult to use because it is not visible" |
| | Legibility/Readability | Enabling the zoom in and out functionalities | 3 | "Being able to zoom in and out of the screen when using a smartphone would be convenient" |
| Directness | Physical accessibility | Adjusting the kiosk height | 6 | "People with visual impairments often look at and select a specific point rather than an entire region, raising the need for aligning the position from which the menu is viewed with the position of the eye" |
| | | Maintaining consistency in the design of the card slot | 6 | "Inconsistency in the design of the card slot makes locating it difficult" |
| | | Avoiding obstruction of access to kiosk passageways | 4 | "In some cases, the kiosk holder protrudes downward, causing people to trip over it. "People with low vision get hurt a lot because they trip over obstacles on the way to their destination" |
| | | Precisely setting the display angle of the kiosk | 2 | "If the kiosk screen is not lying at a particular angle, the contents of the screen are not visible" |

**Table 3.** *Cont.*

| Primary Category | Subcategory | Representative Scenario | Number | Example |
|---|---|---|---|---|
| | User control | Providing plenty of time | 4 | "I would like additional time to use the kiosk" |
| | | Providing sufficient clearance and adequately sized control buttons | 3 | "I have chosen the wrong button from the select and cancel buttons. Since there is no central field of view, it takes a long time to figure out what is being chosen, raising the need for properly spacing the buttons" |
| Efficiency | Effortlessness | Providing guidance for card insertion | 3 | "I am confused about the direction in which the card should be inserted" |

UI: User interface.

**Table 4.** Accessibility items for participants with hearing impairments.

| Primary Category | Subcategory | Representative Scenario | Number | Example |
|---|---|---|---|---|
| User support | Multimodal information | Including non-vocal feedback | 19 | "I feel uncomfortable not having any tools to provide feedback or interact with the kiosk interface to provide inputs" |
| | | Providing sign language interpreters when help button is pressed | 13 | "I pressed the help button at the cafe and called the staff, but I had an uncomfortable experience because the staff could not sign" |
| Directness | User control | Increasing the volume of the sound | 2 | "The sound from the kiosk cannot be clearly heard if the volume of the sound provided by the kiosk is low" |

**Table 5.** Accessibility items for participants with physical impairments.

| Primary Category | Subcategory | Representative Scenario | Number | Example |
|---|---|---|---|---|
| Directness | Physical accessibility | Equipping kiosks with wheelchair accessibility | 27 | "The hardware design of the kiosk makes it difficult to access the front with a wheelchair. Additionally, there is a reduced range of operation when operating a kiosk from the front, raising the need for the wheelchair to be parked sideways to use the kiosk" |
| | | Adjusting kiosk height | 11 | "Regarding kiosks with payment functions, inserting the display downward and placing the card slot sideways would be helpful" |
| | | Adjusting kiosk UI | 5 | "I hope there is a function that can display the UI at the bottom of the kiosk" |

**Table 5.** *Cont.*

| Primary Category | Subcategory | Representative Scenario | Number | Example |
|---|---|---|---|---|
| | User control | Simplifying design of slot for card insertion/removal | 2 | "It is difficult to insert/remove the card at the slot when making a payment. The level of slot is too short, resulting in difficulty in removing hands and dropping movements" |
| | | Diversifying card payment methods | 2 | "Inserting the credit card into the slot can be difficult during payments, raising the need for additional methods such as scratching and tagging" |
| | | Simplifying insertion and removal of hand from the kiosk | 1 | "It is uncomfortable for me to place my hand inside the kiosk to retrieve the receipt when it is issued" |
| | | Incorporating buttons of insufficient sizes | 1 | "The limited button size makes manipulating the controls of the kiosk difficult" |
| Informativeness | Physical visibility | Increasing visibility in top content due to the reflected light from display of the anti-reflective film | 3 | "When looking up from the bottom, the display reflects light, making the top content difficult to see" |
| User support | Privacy | Improving privacy protection | 3 | "I suggest attaching a blocking film to the kiosk so that the operation screen is not visible from the side and placing the entire UI down or displaying important information down to protect it for purposes of privacy" |

## 4.3. Analysis of Usability Items by Impairment Type

Participants with different impairments provided opinions regarding usability when using kiosks. The participants with visual impairments provided many opinions such as "enabling linkage between a smartphone and the kiosk" and "including use of smartphone" (Table 6); those with hearing impairments provided many opinions such as "enabling linkage between a smartphone and the kiosk", "including sign language interpretation service", and "incorporating automated discount for people with disabilities when using the parking lot" (Table 7); and those with physical impairments provided many opinions such as "Making payment system for parking within an arm's reach", "Diversifying card payment methods", and "Incorporating a tablet-ordering-system to improve convenience in ordering"(Table 8).

**Table 6.** Usability items for participants with visual impairments.

| Primary Category | Subcategory | Representative Scenario | Number | Example |
|---|---|---|---|---|
| User support | Effectiveness | Enabling linkage between a smartphone and the kiosk | 5 | "Linking a smartphone and kiosk would do away with the need for a remote control or numeric keypad" |
| | | Reading progressively by voice | 1 | "Reading progressively in order would be simpler than reading each item in the menu one by one" |

**Table 6.** *Cont.*

| Primary Category | Subcategory | Representative Scenario | Number | Example |
|---|---|---|---|---|
| Informativeness | | Equipping kiosks with the ability to recognize disability discounts | 1 | "I require a discount as a person with a disability; there do not seem to be any kiosks that recognize this need" |
| | Visibility | Improving ease in ordering | 1 | "Current voice support is not sufficient for ascertaining the location of the menu" |
| | | Maintaining vertical format of touch screens | 1 | "It would be nice if the touch screen could uniformly follow a vertical format" |
| | Legibility/Readability | Improving the text to picture ratio | 1 | "I prefer text to pictures on the kiosk interface" |
| | Consistency | Maintaining consistency in the usage of the kiosk | 2 | "There should be consistency in how kiosks are used" |
| Learnability | Familiarity | Including use of smartphone | 3 | "Including the smartphone experience can make finding the location of the button easier" |
| | | Including a function to call employees | 5 | "I think I need an employee call function" |
| | Helpfulness | Incorporating helpful guidelines | 1 | "I need guidelines that provide help" |
| | | Providing educational opportunities | 1 | "Educational opportunities should be provided to users who use kiosks for the first time" |
| User support | | Including QR code to clearly explain instructions in manual | 1 | "It would be nice to have a QR code that is easily accessible to explain the device, show the Braille manual, and an enlarged version of the manual" |
| | Error prevention | Including cancellation method for handling situation wherein incorrect icons are clicked | 1 | "I need a way to undo an incorrect action produced by clicking something incorrectly" |
| Simplicity | Modelessness | Fixing menu structure through screen segmentation | 2 | "It would be nice if the menu structure could be fixed by splitting the screen" |
| Directness | User control | Incorporating image/text mode selection function | 2 | "I wish there was an option to choose between either images or text for people with visual impairments" |

QR: Quick response.

**Table 7.** Usability items for participants with hearing impairments.

| Primary Category | Subcategory | Representative Scenario | Number | Example |
|---|---|---|---|---|
| Flexibility | Interoperability | Enabling linkage between a smart phone and the kiosk | 7 | "Providing a personalized service by linking the kiosk with the user's mobile phone" |
| | | Adjusting kiosk height | 3 | "I wish I could adjust the height of the kiosk" |
| | Adaptability | Incorporating individual requirements by impairment type into kiosk design | 1 | "Integrating the requirements of people with different types of disabilities may not guarantee accessibility for all, raising the need to consider each disability individually in the design" |

**Table 7.** *Cont.*

| Primary Category | Subcategory | Representative Scenario | Number | Example |
|---|---|---|---|---|
| Efficiency | Effectiveness | Creating a forum for people with disabilities to receive their opinions and continually make amendments as required | 1 | "The Consumer Agency needs to create a forum for people with disabilities to receive and integrate their opinions and frequently revise them according to the situation" |
| | | Incorporating automated discount for people with disabilities when they use the parking lot | 6 | "In public parking lots, people with disabilities are recognized by their license plates and discounts are provided for them accordingly; this method of providing discounts could be similarly introduced for such people when using kiosks" |
| | | Automatically registering information on people with disabilities and providing them with discount-linked services | 6 | "It would be nice to register the information of people with disabilities that identify their disabilities, such as their fingerprints or welfare cards, to automatically provide them with discount-linked services on places like highways" |
| | | Incorporating a Siren Order system | 2 | "The Siren Order method used at Starbucks is good" |
| | | Incorporating a table ordering system | 1 | "Implementing a table ordering system using a tablet would be good" |
| User Support | Helpfulness | Including sign language interpretation service | 6 | "It would be good to have a sign language interpreter to assist during face-to-face meetings" |
| | | Adding inquiry and consultation method | 3 | "It would be nice to have the contact number of a consultation method that allows communication through text messages in a manner similar to that of KakaoTalk" |
| | | Providing sign language content | 1 | "Providing sign language content to legally ensure that communication is performed" |
| | | Incorporating other means of communication to reduce difficulty in communicating using only call function | 1 | "Since it is difficult to communicate using only the call function of the kiosk, additional communication methods such as text messages, KakaoTalk, and QR codes need to be added" |
| | Error prevention | Including contact means to reduce difficulty in communication in case of errors | 2 | "Not having any means to contact any kiosk technicians in case of an error or incorrect operation such as an inability to retrieve the card from the card slot can be a difficult situation to handle" |
| | Feedback | Clearly displaying timeout sign | 1 | "If a timeout exists, it needs to be clearly displayed on the UI" |
| Informativeness | Comprehensiveness | Providing the same experience at kiosks as that provided by face-to-face employees | 4 | "The experience of ordering using a kiosk should be as seamless as that when ordering face-to-face with an employee" |
| | Consistency | Standardizing kiosk applications | 2 | "It would be convenient if the kiosk app is standardized" |
| | | Including intuitive UI elements | 1 | "Intuitive UI elements such as arrows should be provided on the kiosk screen" |
| | | Providing a consistent way to use kiosk | 1 | "I would like a way that allows the kiosk to be used universally" |

**Table 7.** *Cont.*

| Primary Category | Subcategory | Representative Scenario | Number | Example |
|---|---|---|---|---|
| | Explicitness | Including unique mark on interface for people with hearing impairments | 1 | "People with hearing impairments need a ear-shaped mark rather than a wheelchair-shaped mark on the interface to separate them uniquely from other people" |
| | Visibility | Putting up electronic signboard | 1 | "An electronic sign like that of McDonald's would be good" |
| | Legibility/Readability | Ensuring size of text is reasonable | 1 | "The font on the kiosk screen should be suitably sized" |
| Learnability | Familiarity | Providing Q&As and other important details regarding the kiosk in the instruction manual | 2 | "Information such as Q&As needs to be provided in the manual" |

**Table 8.** Accessibility items for participants with physical impairments.

| Primary Category | Subcategory | Representative Scenario | Number | Example |
|---|---|---|---|---|
| Flexibility | Adaptability | Making payment system for parking within arm's reach | 3 | "When it comes to parking, the payout system should be within an arm's reach" |
| | | Diversifying card payment methods | 2 | "I would like card payment methods such as insertion, scratching, and tagging to be diversified" |
| | Interoperability | Enabling linkage between a smartphone and the kiosk | 1 | "I would like to be able to link my smartphone with the kiosk" |
| Efficiency | Effectiveness | Incorporating a tablet-ordering system to improve convenience in ordering | 2 | "Since each person's arm length is different and some people cannot use their hands, ordering using a tablet would be more convenient" |
| | | Inclusion of car guidance lines to facilitate issuance of parking tickets and card insertion | 1 | "If there are guidelines for parking, it will facilitate the issuance of parking tickets and insertion of cards" |
| | Effortlessness | Increasing convenience in using gift coupons to avoid seeking help from staff | 1 | "Increasing the convenience of using gift coupons would do away with the need to ask for help from the staff" |
| User Support | Helpfulness | Incorporating new means of contact in addition to staff call button | 1 | "When ordering a kiosk, both a phone number and staff call button are needed for seeking assistance during difficulties" |
| Informativeness | Consistency | Improving consistency in payment system | 1 | "I want the payment system to be unified" |

## 5. Accessibility and Usability Characteristics of Kiosks

This study analyzed the accessibility and usability characteristics of kiosks based on the definitions of accessibility and usability as derived through a literature analysis. Based on the categories in Tables 3–8, the concept of accessibility for kiosk devices was divided into the major categories of directness, informativeness, user support, and efficiency, while the concept of usability was divided into the major categories of simplicity, directness, efficiency, informativeness, learnability, user support, and flexibility (Figure 5). Therefore,

when only viewing the major categories analyzed in Tables 1 and 2, accessibility can be seen as a subset of usability, as described in previous studies [50].

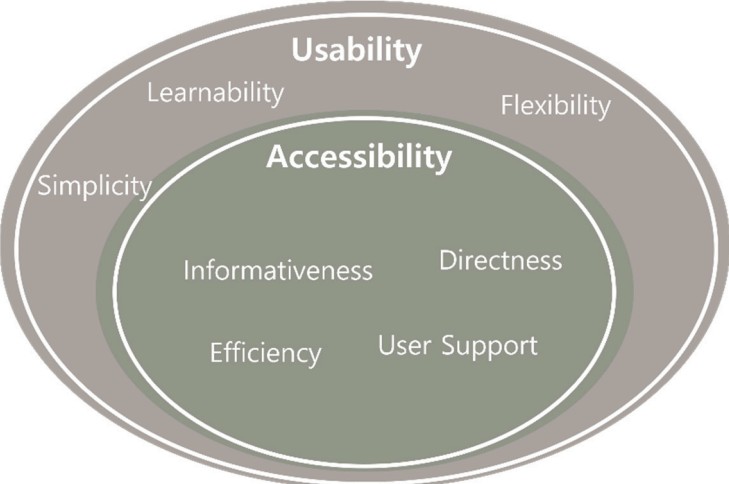

**Figure 5.** Primary categories of accessibility and usability for kiosks.

However, expanding each concept to small categories and examining them thoroughly revealed that despite having certain intersections, accessibility and usability were different concepts. Regarding directness within accessibility, opinions related to the physical accessibility of a kiosk (Tables 3 and 5) and the voice expansion function for using its main functions were derived as representative scenarios (Table 4). In the case of directness within usability, opinions were collected to provide various interface modes for improving the legibility/readability of the text of kiosks (Table 6). Informativeness, physical visibility, visibility, legibility/readability, and consistency were derived for accessibility; visibility, legibility/readability, consistency, comprehensiveness, and explicitness were derived for usability. Informativeness (within usability) had more concepts related to preferred information types (e.g., vertical arrangement information structure and a preference for text over pictures) and content that required more specific and abundant information than the concept of availability/unavailability of information access from a kiosk. User support within accessibility was derived into privacy, multimodal information, feedback, and forgiveness, while user support within usability was derived into feedback, error prevention, and helpfulness; the groups were very clearly different. Finally, efficiency under accessibility was derived to effortlessness related to solving the difficulty of card insertion, whereas the efficiency under usability mostly involved comments related to effectiveness.

We drew a diagram based on the subcategory results of Tables 3–8 related to accessibility and usability (Figure 6). The major category items common to accessibility and usability were expressed by grouping the smaller category items in a circle, while the smaller category items common to both were organized in areas where accessibility and usability overlapped. For example, informativeness is a primary category common to accessibility and usability, and legibility/readability, visibility, and consistency are the small categories common to them. In the case of flexibility, the primary categories of the items were organized in the usability area because they only existed there. Although accessibility and usability had common denominators in legibility/readability, visibility, consistency, and user control, the analysis of the subcategories confirmed that they were very different concepts. Additionally, many opinions on accessibility involved essential functions, while many on usability involved additional functions and services; psychological factors such as personal preference were also involved.

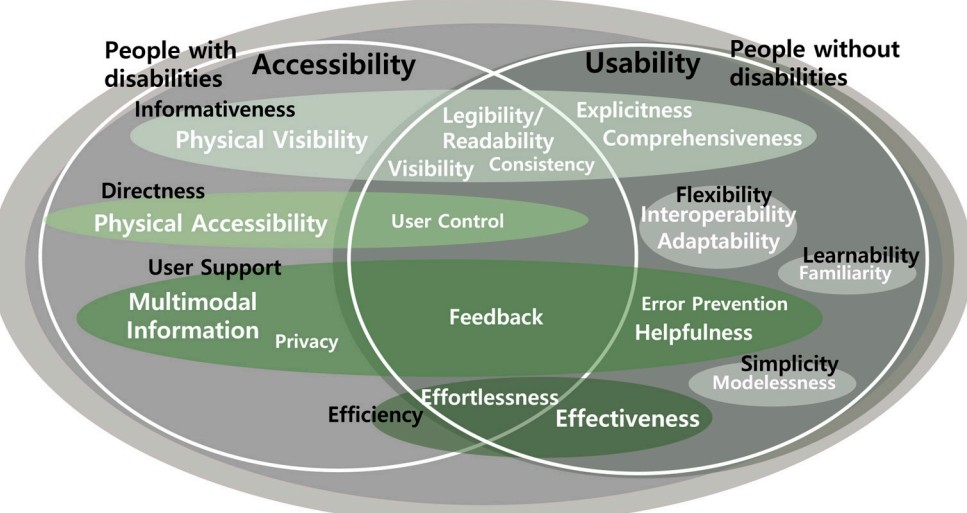

**Figure 6.** Mapping of categories of accessibility and usability for kiosks.

## 6. Limitations

### 6.1. Participants and Devices

The focus of this study was on the accessibility and usability of kiosks. Thus, the subclass factors included in Figure 6 may vary for other devices or services. For example, because a smartphone or tablet is portable, physical accessibility may be excluded as a factor. Additionally, because the study mainly involved people with disabilities, the concept of accessibility and usability may be somewhat different for the elderly.

A total of 31 respondents participated in this study, and it may be considered that the number is not sufficient. However, according to a previous study, it is known that in the case of a 2-hour FGI, if there are eight or more respondents, more than 90% of user needs can be identified [51]. In fact, in previous accessibility-related studies, it is common for the number of participants to be less than 20 [52,53]. Therefore, the sample of this study is not representative enough, but the experiences derived from Tables 3–8 are thought to include more than 90% of the kiosk experience.

### 6.2. Methodology of Focus Group Interview

Because FGIs induce participants to share their opinions mainly through verbal communication, the amount of information that was derived from hearing-impaired participants may have been insufficient. Owing to the large volume of opinions shared, it was difficult to assess the effectiveness of the main moderator facilitating the communication of participants with hearing impairments through sign language.

Finally, because past memories were used to evaluate the kiosk experience [54], the participants may have provided opinions that were less detailed than the ones they had when using the kiosk directly. However, considering there are no kiosks that people with visual impairments can use, this method can be concluded as the most effective for gathering information from participants.

## 7. Conclusions

The purpose of this study was to define the concepts of accessibility and usability for kiosks and identify the characteristics of these concepts for people with different types of disabilities. First, the concepts of accessibility and usability were investigated through a literature review. Then, FGIs were conducted to collect comprehensive experiences on the accessibility and usability of kiosks for some participants and organize the characteristics of these concepts accordingly. Based on the collected data, we have presented a diagram that analyzes the relationship between accessibility and usability.

The opinions collected through the FGIs were classified into 78 items, 33 of which were classified as accessibility and 45 of which were classified as usability. The derived items were mapped according to previously organized criteria for accessibility and usability, and the number of opinions for each item was derived. The items for accessibility were derived in the order of visual impairment, physical disability, and hearing impairment, while those for usability were derived in the order of hearing impairment, visual impairment, and physical disability. The concepts of accessibility and usability were defined based on the opinions derived through the FGIs. Accessibility mainly had characteristics related to essential functions, and usability mainly had characteristics related to additional functions. The detailed definitions of accessibility and usability for kiosks in this study are expected to be useful in developing guidelines and laws for using kiosks in the future.

**Author Contributions:** Conceptualization, H.K.K.; methodology, H.K.K.; validation, Y.L., S.P., J.P. and H.K.K.; investigation, Y.L. and H.K.K.; writing—original draft preparation, H.K.K. and Y.L.; writing—review and editing, H.K.K. and J.P.; All authors have read and agreed to the published version of the manuscript.

**Funding:** This work was supported by a National Research Foundation of Korea (NRF) grant funded by the Korean government (MSIT) (No. NRF-2021R1F1A1063155).

**Institutional Review Board Statement:** The study was conducted in accordance with the Declaration of Helsinki and was approved by the Institutional Review Board of Kwangwoon University (IRB approval: 2020-08-005).

**Informed Consent Statement:** Informed consent was obtained from all subjects involved in the study.

**Data Availability Statement:** The data presented in this study are available on request from the corresponding author.

**Conflicts of Interest:** The authors declare no conflict of interest.

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
