# Peer review of "Comparative Analysis of Usability and Accessibility of Kiosks for People with Disabilities"

_applsci, doi:10.3390/app13053058_

Round 1

Reviewer 1 Report

Please enlist the criteria followed by the user while recruiting the participants.

On what quantitative metrics, the contribution of all the parameters is detected in Figure 5 and Figure 6?

Author Response

Please enlist the criteria followed by the user while recruiting the participants.

Thank you for your valuable comments. We have added the following sentence in session 3.1 by stating “The participants were recruited from among users with visual, hearing, and physical disabilities, who had experience using a kiosk or were interested in using it.”

On what quantitative metrics, the contribution of all the parameters is detected in Figure 5 and Figure 6?

Thank you for your valuable comments. We have added the following sentences in session 5 by stating “Based on the categories in Tables 3 to 8, the concept of accessibility for kiosk devices was divided into the major categories of directness, informativeness, user support, and efficiency while the concept of usability was divided into the major categories of simplicity, directness, efficiency, informativeness, learnability, user support, and flexibility (Figure 5).”, and We drew a diagram based on the subcategory results of Table 3 to 8 related to accessibility and usability (Figure 6).”.

Reviewer 2 Report

The paper isdevouted Usability and Accessibility of Kiosks for People With Visual, Hearing, and Physical Impairments Using Focus Group Interviews-

For scientific method was choosen comparative analysis.

The aim or article is sounds good. But there are some remarks

1. The title of article have to be shorted.

2. The sample is not represantative enought. It needs inincreasing of the number of respondents. 

3. Additional checks for the validity of the results were not performed.

4. The Referemces needs in improving. Authors have to include scientific papers not older then 5 years (aproximatly 70 % of all litarary sourses.

So oweral merit is that the paper is needed in major improovment (Reconsider after major revision).

Author Response

  1. The title of article have to be shorted.

Thanks for the good comments. The title has been changed as follows; “Comparative Analysis of Usability and Accessibility of Kiosks for People with Disabilities”

  1. The sample is not represantative enought. It needs inincreasing of the number of respondents. 

Thanks for the good comments. The sample is about 10 people for each type of disability, and we know that it is not representative enough. However, there is a previous study that found that about 90% of usability problems can be derived if more than 8 participants participate in FGI [48]. In addition, it is common to have around 20 people in previous accessibility FGI studies [45,50].

We added the limitation of this study in session 6.1 by stating ““A total of 31 respondents participated in this study, and it may be considered that the number is not sufficient. However, according to a previous study, it is known that in the case of a 2-hour FGI, if there are 8 or more respondents, more than 90% of user needs can be identified [48]. In fact, in previous accessibility-related studies, it is common for the number of participants to be less than 20 [45, 49]. Therefore, the sample of this study is not representative enough, but the experiences derived from Tables 3 to 8 are thought to include more than 90% of the kiosk experience."

  1. Kim, H. K.; Jeong, H.; Park, J.; Park, J.; Kim, W. S.; Kim, N.; Park, S.; Paik, N. J. Development of a comprehensive design guideline to evaluate the user experiences of meal-assistance robots considering human-machine social interactions. 2022. Int. J. Hum–Comput. Int, 38(17), 1687-1700; DOI:10.1080/10447318.2021.2009672
  2. Griffin, Abbie and John R. Hauser. “The Voice of the Customer”, Marketing Science. vol. 12, no. 1, Winter 1993.
  3. Barrett, J.,; Kirk, S. Running focus groups with elderly and disabled elderly participants. 2020. Applied ergonomics31(6), 621-629; DOI: 10.1016/S0003-6870(00)00031-4

  1. Additional checks for the validity of the results were not performed.

The authors understand the reviewers' concerns about the validity of the results. In this qualitative analysis, data analysis must be conducted systematically for validity, and this study also tried to do this. However, the current paper lacks an explanation of the relevant part, so the following sentences have been added by stating in session 3. “This qualitative classification was conducted by four accessibility experts, who individually categorized participants’ opinions into the categories and subcategories of usability and accessibility. The principle of classification was that all four experts map to the same categories or subcategories. If even one expert had opinions mapped to different categories or subcategories, a discussion was conducted so that all four experts could derive the same classification result.”

  1. The Referemces needs in improving. Authors have to include scientific papers not older then 5 years (aproximatly 70 % of all litarary sourses.

Reflecting the opinions, the following latest references were added to the paper.

  1. Akgül, Y. Accessibility, usability, quality performance, and readability evaluation of university websites of Turkey: a comparative study of state and private universities. Univ. Access Inf. Soc., 20(1), 157-170; DOI:10.1006/ijhc.1998.0243.
  2. MacakoÄŸlu, Åž. S.; Peker, S.; Medeni, İ. T. Accessibility, usability, and security evaluation of universities’ prospective student web pages: a comparative study of Europe, North America, and Oceania. Univ. Access Inf. Soc., 1-13; DOI:10.1007/s10209-022-00869-9
  3. Kim, H. K.; Jeong, H.; Park, J.; Park, J.; Kim, W. S.; Kim, N.; Park, S.; Paik, N. J. Development of a comprehensive design guideline to evaluate the user experiences of meal-assistance robots considering human-machine social interactions. Int. J. Hum–Comput. Int, 38(17), 1687-1700; DOI:10.1080/10447318.2021.2009672
  4. Csontos, B.; Heckl, I. Accessibility, usability, and security evaluation of Hungarian government websites. Univ. Access Inf. Soc., 20, 139-156; DOI:10.1007/s10209-020-00716-9
  5. Ismail, A.; Kuppusamy, K.S.; Paiva, S. Accessibility analysis of higher education institution websites of Portugal. 2020. Access Inf. Soc., 19, 685–700; DOI: 10.1007/s10209-019-00653-2

Reviewer 3 Report

The submission defined the concepts of accessibility and usability for kiosks and identify the characteristics of these concepts for people with different types of disabilities, and presented a diagram to analyzes the relationship between accessibility and usability. The paper is interesting, and can be useful for creating laws and guidelines regarding the accessibility and usability of kiosks.

Author Response

The submission defined the concepts of accessibility and usability for kiosks and identify the characteristics of these concepts for people with different types of disabilities, and presented a diagram to analyzes the relationship between accessibility and usability. The paper is interesting, and can be useful for creating laws and guidelines regarding the accessibility and usability of kiosks.

Thank you for your good review comments.

Round 2

Reviewer 2 Report

The authors corrected the comments made, although not completely.

But in this form, the article is more suitable for publication and can be аccepted after minor revision.

If the authors could improve the quality, taking into account the previous comments, this would significantly improve the scientific level of the publication.

Author Response

Thank you for your valuable comments. Perhaps it is necessary to supplement number 4 of the previous comments. which was "The References needs in improving. Authors have to include scientific papers not older then 5 years (approximately 70 % of all literary sources."

Accordingly, the research team modified and added the following numbers to include the latest research (2, 4, 13, 18, 26, 27, 30, 36, 37, 38, 41, 42, 43, 44, 45 , 46, 47, 49, 50, 51, 52, 53, 54). Finally, a total of 34 studies after 2017 were referenced in the revised paper (1, 2, 4, 6, 11, 13, 14, 15, 18, 23, 24, 25, 26, 27, 30, 35, 36, 37, 38, 40, 41, 42, 43, 44, 45, 46, 47, 49, 50, 51, 52, 53, 54), accounting for about 63% of the total number of references.

Below are the revised/added references. Thanks to the good comments, the quality of the thesis could be improved.

2. Lazar, J.; Jordan, J. B.; Vanderheiden, G. Toward unified guidelines for kiosk accessibility. Interactions. 2019, 26(4), 74-77.

4. Caporusso, N.; Udenze, K.; Imaji, A.; Cui, Y.; Li, Y.; Romeiser, S.. Accessibility evaluation of automated vending machines. In Advances in Design for Inclusion: Proceedings of the AHFE 2019 International Conference on Design for Inclusion and the AHFE 2019 International Conference on Human Factors for Apparel and Textile Engineering, July 24-28, 2019, Washington DC, USA 10 (pp. 47-56). Springer International Publishing.

13. Alajarmeh, N. Evaluating the accessibility of public health websites: an exploratory cross-country study. Univ. Access Inf. Soc. 2022, 21(3), 771-789; DOI: 10.1007/s10209-020-00788-7

18. Beyene, W. M.; Mekonnen, A. T.; Giannoumis, G. A. Inclusion, access, and accessibility of educational resources in higher education institutions: exploring the Ethiopian context. International Journal of Inclusive Education. 2023, 27(1), 18-34; DOI: 10.1080/13603116.2020.1817580

26. Kim, N.; Park, J.; Park, J.; Kim, H. K.; Choe, M.; Park, J.; Kim, J. Usability evaluation of symbols in digital cluster for drivers with color vision deficiency. Univ. Access Inf. Soc. 2020, 1-15. DOI: doi.org/10.1007/s10209-022-00898-4

27. Gómez-Carmona, O.; Casado-Mansilla, D.; López-de-Ipiña, D. Multifunctional interactive furniture for smart cities. In Proceedings (Vol. 2, No. 19, p. 1212). November 2018.

30. Pacheco, P.; Santos, F.; Coimbra, J.; Oliveira, E.; Rodrigues, N. F. Designing Effective User Interface Experiences for a Self-Service Kiosk to Reduce Emergency Department Crowding. In 2020 IEEE 8th International Conference on Serious Games and Applications for Health (SeGAH), 1-8, August 2020.

36. Buß, R. Inclusive Design–Go Beyond Accessibility. In Human-Computer Interaction. Human Values and Quality of Life: Thematic Area, HCI 2020, Held as Part of the 22nd International Conference, HCII 2020, Copenhagen, Denmark, July 19–24, 2020, Proceedings, Part III 22 (pp. 400-407). Springer International Publishing.

37. Moore, A.; Boyle, B.; Lynch, H. Designing for inclusion in public playgrounds: a scoping review of definitions, and uti-lization of universal design. Disability and Rehabilitation: Assistive Technology, 2022, 1-13; DOI: 10.1080/17483107.2021.2022788

38. Silva, G. M. S.; de C. Andrade, R. M.; de Gois R. Darin, T. Design and evaluation of mobile applications for people with visual impairments: a compilation of usable accessibility guidelines. In Proceedings of the 18th Brazilian Symposium on Human Factors in Computing Systems, 1-10 October 2019.

41. Vakulenko, Y.; Oghazi, P.; Hellström, D. Innovative framework for self-service kiosks: Integrating customer value knowledge. Journal of Innovation & Knowledge. 2019, 4(4), 262-268; DOI: 10.1016/j.jik.2019.06.001

42. Murray, M. Narrative data. In U. Flick (Ed.), The SAGE handbook of qualitative data collection (pp. 264-280). London, England: Sage. 2018.

43. Akgül, Y. Accessibility, usability, quality performance, and readability evaluation of university websites of Turkey: a comparative study of state and private universities. Univ. Access Inf. Soc., 2021. 20(1), 157-170; DOI:10.1006/ijhc.1998.0243.

44. MacakoÄŸlu, Åž. S.; Peker, S.; Medeni, İ. T. Accessibility, usability, and security evaluation of universities’ prospective student web pages: a comparative study of Europe, North America, and Oceania. 2022. Univ. Access Inf. Soc., 1-13; DOI:10.1007/s10209-022-00869-9

45. Kim, H. K.; Jeong, H.; Park, J.; Park, J.; Kim, W. S.; Kim, N.; Park, S.; Paik, N. J. Development of a comprehensive design guideline to evaluate the user experiences of meal-assistance robots considering human-machine social interactions. 2022. Int. J. Hum–Comput. Int, 38(17), 1687-1700; DOI:10.1080/10447318.2021.2009672

46. Csontos, B.; Heckl, I. Accessibility, usability, and security evaluation of Hungarian government websites. 2021. Univ. Access Inf. Soc., 20, 139-156; DOI:10.1007/s10209-020-00716-9

47. Ismail, A.; Kuppusamy, K.S.; Paiva, S. Accessibility analysis of higher education institution websites of Portugal. 2020. Univ. Access Inf. Soc., 19, 685–700; DOI: 10.1007/s10209-019-00653-2

49. Barrett, J.; Kirk, S. Running focus groups with elderly and disabled elderly participants. 2020. Applied ergonomics, 31(6), 621-629; DOI: 10.1016/S0003-6870(00)00031-4

50. Weichbroth, P. Usability of mobile applications: a systematic literature study. IEEE ACCESS. 2000. 8, 55563-55577; DOI: 10.1109/ACCESS.2020.2981892

51. Borsci, S.; Federici, S.; Malizia, A.; De Filippis, M. L. Shaking the usability tree: why usability is not a dead end, and a constructive way forward. Behav. Inf. Technol. 2019. 38(5), 519-532; DOI: 10.1080/0144929X.2018.1541255

52. Coyle, N.; Kennedy, A.; Schull, M. J.; Kiss, A.; Hefferon, D.; Sinclair, P.; Alsharafi, Z. The use of a self-check-in kiosk for early patient identification and queuing in the emergency department. Canadian Journal of Emergency Medicine. 2019, 21(6), 789-792; DOI: 10.1017/cem.2019.349

53. Van De Sanden, S.; Willems, K.; Brengman, M. How customers motive attributions impact intentions to use an interactive kiosk in-store. Journal of Retailing and Consumer Services. 2022, 66, 102918; DOI: 10.1016/j.jretconser.2022.102918

54. Nam, J.; Kim, S. Why do elderly people feel negative about the use of self-service technology and how do they cope with the negative emotions?. 2022. 31st European Conference of the International Telecommunications Society (ITS): "Reining in Digital Platforms? Challenging monopolies, promoting competition and developing regulatory regimes", Gothen-burg, Sweden, 20th - 21st June 2022